# Long-Term Neurodevelopmental Outcome in Twin-to-Twin Transfusion Syndrome: Is there still Room for Improvement?

**DOI:** 10.3390/jcm8081226

**Published:** 2019-08-15

**Authors:** Marjolijn S. Spruijt, Enrico Lopriore, Ratna N.G.B. Tan, Femke Slaghekke, Frans J.C.M. Klumper, Johanna M. Middeldorp, Monique C. Haak, Dick Oepkes, Monique Rijken, Jeanine M.M. van Klink

**Affiliations:** 1Department of Pediatrics, Division of Neonatology, Leiden University Medical Center, J7-48, Albinusdreef 2, 2333 ZA Leiden, The Netherlands; 2Department of Obstetrics, Division of Fetal Therapy, Leiden University Medical Center, Albinusdreef 2, 2333 ZA Leiden, The Netherlands

**Keywords:** twin-to-twin transfusion syndrome, long-term outcome, neurodevelopmental impairment, fetoscopic laser surgery, risk factors

## Abstract

Despite many developments in its management, twin-to-twin transfusion syndrome (TTTS) remains an important risk factor for long-term neurodevelopmental impairment (NDI). Our objective was to compare the incidence of severe NDI in a recent cohort of TTTS survivors, treated with laser surgery from 2011 to 2014, with a previous cohort treated from 2008 to 2010. Neurological, cognitive, and motor development were assessed at two years of age. We determined risk factors associated with Bayley-III scores. Severe NDI occurred in 7/241 (3%) survivors in the new cohort compared to 10/169 (6%) in the previous cohort (*p* = 0.189). Disease-free survival (survival without severe impairment) did not significantly differ. Low birth weight and being small for gestational age (SGA) were independently associated with lower cognitive scores (both *p* < 0.01). Severe cerebral injury was related to decreased motor scores (B = −14.10; 95% CI −3.16, −25.04; *p* = 0.012). Children with severe NDI were born ≥32 weeks’ gestation in 53% of cases and had no evidence of cerebral injury on cranial ultrasound in 59% of cases. Our results suggest that improvement in outcome of TTTS has reached a plateau. Low birth weight, SGA, and cerebral injury are risk factors for poor neurodevelopmental outcome. Neither gestational age above 32 weeks nor the absence of cerebral injury preclude severe NDI.

## 1. Introduction

Monochorionic twin pregnancies complicated by twin-to-twin transfusion syndrome (TTTS) carry a high risk of fetal and neonatal complications, including death and long-term neurodevelopmental impairment (NDI) [1,2]. Fetoscopic laser coagulation of the causative placental vascular anastomoses has proven to be superior to serial amnioreduction in the treatment of TTTS [3,4,5,6,7]. Laser surgery has been the standard treatment for TTTS in the Leiden University Medical Center (LUMC) since 2000. Because of the invasive nature of the treatment for this high-risk disease, we have since then advocated that long-term follow-up of surviving infants should be an integral part of any fetal therapy program. 

In a previous study, we showed a marked improvement in overall survival in TTTS treated at our center, from 70% in a cohort of pregnancies treated in the first six years of our laser surgery program (2000–2005), to 80% in a cohort treated between 2008 and 2010. Improved survival was associated with a decrease in severe NDI from 18% to 6% [8,9]. This may be explained by the growing experience of our fetal therapy team, leading to improved monitoring and management, as well as a learning-curve effect of the highly technical fetoscopic laser procedure [10]. Other authors have also suggested that there is a decreasing trend in the incidence of severe NDI in more recent studies [11].

The purpose of the current study was to evaluate if neurodevelopmental outcome has continued to improve over time. We assessed neurodevelopment in TTTS survivors treated with laser surgery at our center between 2011 and 2014 and compared these results with a previous cohort treated between 2008 and 2010. In addition, we aimed to identify predictors for adverse neurodevelopmental outcome after laser treatment for TTTS using follow-up data from all children treated between 2008 and 2014. 

## 2. Experimental Section

The LUMC is the national referral center for TTTS in the Netherlands. For this study, we included all consecutive TTTS pregnancies treated with laser surgery in the LUMC between 1 January 2011 and 31 December 2014 (new cohort) and compared the neurodevelopmental outcome of surviving children with an earlier cohort, treated between 1 January 2008 and 31 December 2010 (previous cohort). The 2-year outcome of a large part of this 2008 to 2010 cohort has been published previously [9]. TTTS diagnosis and staging were determined with ultrasound criteria according to Quintero [12,13]. During the two study periods, the selective laser technique and the Solomon technique, which was studied in the Solomon trial that ran from March 2008 until July 2012, were used for the treatment of TTTS [14]. After conclusion of the trial, the Solomon technique became the standard technique for all laser procedures.

Spontaneous monoamniotic pregnancies, children with major congenital anomalies, and children with other causes of severe developmental delay unrelated to TTTS were excluded from this study. For each TTTS pregnancy, we recorded the following data from the medical records: gestational age (GA) at laser surgery, TTTS stage, fetal demise, delivery <24 weeks gestation, selective cord coagulation after laser therapy, occurrence of twin anemia polycythemia sequence (TAPS) after laser, and recurrence of TTTS after laser. Fetal demise included post-laser demise and death due to birth before 24 weeks gestation, the legal limit for neonatal resuscitation in the Netherlands. For each neonate, the following data were recorded: former donor or recipient status, gender, GA at birth, birth weight, whether or not small for gestational age (SGA, defined as birth weight below the 10th percentile), presence of severe cerebral injury, and neonatal death. TAPS was diagnosed according to antenatal and/or postnatal criteria [15]. Severe cerebral injury was defined as the presence of at least one of the following findings on cerebral ultrasound: intraventricular hemorrhage (IVH) ≥ grade III, periventricular leukomalacia (PVL) ≥ grade II, ventricular dilatation ≥97th percentile, and porencephalic cyst, arterial infarction, venous hemorrhagic infarction, or other severe cerebral lesions associated with adverse neurological outcome [16]. Neonatal cranial ultrasonography was performed by experienced neonatologists in accordance with our clinical protocol, as described in detail in our earlier studies [17]. Our protocol states that TTTS is an indication for at least one cranial ultrasound in the first week of life, independent of GA at birth. Neonatal death was defined as the death of a neonate born at or after 24 weeks 0 days of gestation, within 28 days after birth.

All TTTS survivors were invited for follow-up assessment at two years of age, corrected for prematurity. This visit consisted of a neurological and physical examination by a pediatrician and physiotherapist. When cerebral palsy (CP) was diagnosed, it was defined according to the European CP Network and classified using the Gross Motor Function Classification System (GMFCS) [18]. Additionally, cognitive and motor development were evaluated by a child psychologist using the Bayley Scales of Infant and Toddler Development-third edition (Bayley-III) [19]. The results of the Bayley-III are expressed as cognitive and motor composite scores. Each of these scores has a normal distribution with a mean of 100 and standard deviation of 15. Until the Dutch version (Bayley-III-NL) became available in 2015, children were tested with the United States (US) version. The Bayley-III-NL is the translated and slightly adapted version of the original Bayley-III. For this study, the raw US test results were re-evaluated according to Dutch norms, which resulted in slightly different motor and cognitive composite scores. These ‘Dutch’ composite scores were used for all analyses in this study.

We determined our primary outcome, called severe neurodevelopmental impairment (NDI), as any of the following: CP GMFCS grade >1, cognitive or motor composite score <70, bilateral blindness or bilateral deafness requiring amplification. The primary outcome was determined in children when at least the visit with the physician and physiotherapist and the cognitive subtest of the Bayley-III were completed. 

For this study, we determined the rate of disease-free survival, defined as the number of children who were free of NDI at two years adjusted age divided by the total number of included fetuses for whom the outcome was known, that is, either fetal/neonatal death or the presence or absence of severe NDI. Severe NDI and disease-free survival were compared between the two cohorts. For the recent cohort, we also evaluated the incidence of mild NDI, defined as either CP GMFCS grade 1, or Bayley-III cognitive or motor composite score ≥70 and <85.

The long-term outcome data from both cohorts together were analyzed to determine which risk factors influence cognitive and motor scores of the Bayley-III.

No formal ethical approval was required for this anonymized retrospective study. The institutional review board of the LUMC reviewed the study protocol and declared a statement of no objection.

### Statistical Analysis

Results are reported as percentages, mean and SD or median and range, as appropriate. A *p*-value below 0.05 was considered to be statistically significant. Analyses were performed using generalized estimating equations (GEE), to account for the fact that results are not independent within twin pairs. The relationship between potential risk factors with Bayley-III scores was investigated with a linear regression analysis using the GEE approach. The following risk factors were studied: former donor or recipient status, TTTS stage, GA at laser, incomplete laser (either TAPS or recurrent TTTS after laser), fetal demise of the co-twin, GA at birth, birth weight, SGA, and severe cerebral injury. Significant factors in the univariate analysis were included in a multivariate model to estimate the independent effects. Results of the GEE risk factor analysis are expressed as regression coefficient B with 95% confidence interval (95% CI). Our data were analyzed using IBM SPSS Statistics, Version 23.0; Chicago, IL, USA.

## 3. Results

### 3.1. Perinatal Characteristics of Included TTTS Pregnancies 

Between January 2011 and December 2014, all consecutive 204 TTTS pregnancies treated with laser at the LUMC were included in this follow-up study. We compared the outcome of this new cohort with 116 TTTS pregnancies from the previous cohort treated between 2008 and 2010. This adds up to a total of 320 TTTS pregnancies, or 640 fetuses, treated with laser surgery in seven years’ time. A flow chart showing the total study population divided into the two cohorts is depicted in Figure 1.

There were no significant differences in any of the perinatal characteristics between the two cohorts. Fetoscopic laser surgery took place at a mean GA of 19.6 ± 3.3 weeks in our most recent cohort. Median Quintero stage at time of treatment was 3 in both cohorts. Post-laser TAPS and recurrent TTTS were seen in 11% and 10% of pregnancies in the two cohorts, respectively. The incidences of fetal demise in the new and previous cohort were 19% and 15%, respectively (*p* = 0.163). For our most recent cohort, we determined the type of fetal demise. In total, fetal demise occurred in 79/408 (19%) of cases, of which 39 single and 40 double fetal demises (i.e., 20 twin pairs). In 52/79 (66%) cases, fetal demise occurred in utero after laser therapy. In 12 pregnancies, or 24/79 fetuses (30%), spontaneous delivery took place before 24 weeks of gestation, which is the legal limit for active neonatal resuscitation in the Netherlands. Mean GA at delivery in these pregnancies was 20.8 weeks (range 17–23 weeks). The remaining 3/79 (4%) died after selective cord coagulation because of complications detected after laser surgery (one with severe cerebral injury, one with severe growth restriction, and one with TAPS after incomplete laser). Mean GA at live birth was 32.4 weeks and mean birth weight 1742 grams in the new cohort, both similar to the previous cohort. The incidences of severe cerebral injury were 6% in the new cohort and 7% in the previous one. The rate of neonatal death also remained stable over time and occurred in 6% (new cohort) versus 5% (previous cohort). Overall, 309/408 (76%) of treated TTTS fetuses survived to at least 28 days after birth, which is not significantly different from the previous cohort (81%, *p* = 0.159).

### 3.2. Long-Term Follow-Up

In our most recent cohort, 47/309 (15%) survivors were lost to follow-up. Four survivors were excluded from analysis of long-term outcome because of other causes of neurodevelopmental delay. One twin pair was diagnosed with neurofibromatosis type 1. Another twin pair had severe, familial, bilateral hearing loss, and a genetic cause was suspected. Follow-up data were available for the remaining 258/309 (84%) survivors. Table 1 shows the baseline characteristics of all TTTS survivors included for follow-up. No differences between the cohorts were found.

The perinatal characteristics of the included versus the lost-to-follow-up group in our most recent cohort were compared. Mean GA at birth in the follow-up group was higher compared to the lost-to-follow-up group (32.9 weeks vs. 31.7 weeks, *p* = 0.039). For all other characteristics, the groups were similar.

#### Neurodevelopmental Outcome

The results of the follow-up assessments at the corrected age of two years are presented in Table 2. The incidences of severe NDI, CP, and disease-free survival were not significantly different in the new compared to the previous cohort. There was no difference in mean Bayley-III cognitive scores between the new and previous cohort, whereas mean motor composite score in the new cohort was significantly lower compared to the previous cohort.

In the new cohort, follow-up was complete in 232/258 (90%) children. Out of 258 children, 8 completed only the cognitive subtest and 2/258 children only the motor subtests of the Bayley-III. The presence or absence of severe NDI could be determined in 241 children: 240 children who completed at least the cognitive Bayley-III, and 1 child without a Bayley-III test who was bilaterally deaf. In the previous cohort, follow-up was complete in 164/176 (93%) children, and in 5/176 children, only the cognitive subtest of the Bayley-III was completed. The presence or absence of severe NDI could be determined in these 169 children.

For all children in the previous cohort and 142/242 children in the new cohort, the original Bayley-III was used, and the raw scores were re-evaluated according to the Dutch norms for this study. For the other 100/242 children in the new cohort, the Bayley-III-NL was used. 

We did not observe blindness in any of our follow-up patients. Bilateral deafness was present in one former recipient (0.4%) in our most recent cohort.

Mild NDI was assessed in the new cohort and was present in 53/232 (23%) children with a complete Bayley-III. A cognitive score between 70 and 84 was found in 28/240 (12%). A motor score between 70 and 84 was found in 47/234 (20%).

Of the total of 17 children with severe NDI, 9 (53%) were born at gestational ages ≥32 0/7 weeks, and 10 (59%) had no evidence of severe cerebral injury on cranial ultrasound performed during the neonatal period. Six children (35%) fell into both categories (i.e., they were >32 weeks without evidence of severe cerebral damage).

### 3.3. Risk Factors

Results of the univariate linear regression analysis of potential risk factors for all survivors from the two cohorts taken together are shown in Table 3. It shows a significant association between birth weight and cognitive scores: for each 100 grams increase in birth weight, cognitive scores increased with 0.41 point (B 0.41; 95% CI 0.18−0.64, *p* = 0.000). We found a trend towards an association between higher GA at birth and higher cognitive scores (B 0.49; 95% CI −0.01 to 0.99, *p* = 0.056). There was a strong positive correlation between GA at birth and birth weight (r = 0.86; *p* = 0.01). In addition, children who were SGA had significantly lower cognitive scores than children with birth weights above the 10th percentile for GA (B −5.67; 95% CI −9.35 to −1.99, *p* = 0.003). In a multivariate analysis, both birth weight (B 0.34; 95% CI 0.11–0.57, *p* = 0.004) and SGA (B −4.28; 95% CI −7.91 to −0.65, *p* = 0.021) remained independently associated with cognitive scores (not shown in table). For motor scores, univariate analysis revealed a significant negative association with severe cerebral injury, leading to motor scores that were 14.10 points lower in children with severe cerebral injury compared to children without severe cerebral injury (B −14.10; 95% CI −25.04 to −3.16, *p* = 0.012).

## 4. Discussion

This is the largest study describing neurodevelopmental outcome of TTTS survivors published to date, studying over 600 fetuses, of which 434 children were available for follow-up at two years of age. Overall survival was 76% in the cohort treated between 2011 and 2014, which was similar to 81% survival in the previous cohort.

In our most recent cohort, the incidences of severe NDI and CP were 3% and 2%, respectively. Earlier follow-up studies of laser-treated TTTS twins, including our own, have reported incidences of severe NDI between 6% and 18% and of CP between 3% and 11%, with a trend toward lower incidences of NDI in more recent studies [9,11,20]. We found no significant differences in severe NDI or disease-free survival between the two consecutive cohorts in the current study. The low numbers of children with severe NDI and CP presumably contribute to the lack of significance. However, the fact that disease-free survival has not increased further seems to indicate that the considerable improvement in outcome for TTTS that was achieved in the first ten years of our laser surgery program has now reached a plateau. In a devastating disease such as TTTS, bringing disease-free survival up to the level of the general population is probably not feasible. In addition to the remaining risk of severe NDI due to the severe nature of the disease, important limiting factors for further improvement of disease-free survival are intrauterine fetal demise and spontaneous premature delivery after laser surgery. Together, fetal demise and birth before the threshold of viability occurred in 19% of fetuses in our latest cohort. Finding ways to reduce these risks may contribute to further improvement of disease-free survival after laser treatment for TTTS. 

In this study, we found a significantly lower mean motor score in our new cohort compared to the previous cohort. A possible explanation for the difference in motor score is that different child psychologists have tested the children since 2000. Although the inter-observer agreement for the Bayley-III is substantial with a kappa coefficient of 0.77, an individual tendency to score a certain way may have an effect on the average results [19]. Another possible explanation for this difference is that about half of the children in the new cohort were tested with the Bayley-III-NL, whereas the rest of the children were tested with the original US version of the Bayley-III. In the Bayley-III-NL, the reversal rule of the gross motor subtest was adapted and became stricter [21]. This adaptation may have contributed to lower motor scores in the children tested with the Bayley-III-NL, which is in favor of the previous cohort. The decrease in motor score that was found in this study may also reflect a true, unexplained effect. 

In the current study, we chose to also report mild NDI for several reasons. First, the incidence of severe NDI has become lower over the years, shifting attention to children with mild impairments, which can still have a major impact on the care and educational requirements of children. Secondly, several studies have shown that test scores of the Bayley-III are higher when compared to the previous version (Bayley-II), resulting in the risk of underestimation of NDI with the use of the Bayley-III [22]. Our study shows that the incidence of mild NDI in the new cohort was considerable, as 23% of children fell into this category. Data on mild—sometimes called minor or moderate—NDI are limited, but it was present in 11% of children in a study that used the Griffiths Scales of Child Development 2nd edition, and in respectively 7% and 29% of children in two studies that used the Ages and Stages Questionnaire (ASQ) [23,24,25]. Due to different testing methods and definitions of mild NDI, a direct comparison cannot be made.

Cerebral injury in TTTS is thought to occur at different stages and may be caused by fetal hemodynamic imbalance, anemia or polycythemia before laser surgery. Sudden changes in hemodynamics during the procedure may also cause ischemic or hemorrhagic injury to the fetal brain. The association between birth weight and growth restriction with cognitive performance found in our study has been described in several other studies [26,27]. However, in contrast with other studies, the association between GA at birth and neurodevelopment in this large group of 434 children was not significant (*p* = 0.056). In fact, more than half of the children with severe NDI were born at gestational ages of 32 weeks or more. The overall improvement in neonatal intensive care treatment at younger gestational ages, in combination with the low absolute number of TTTS survivors with severe NDI, may cause the association between lower gestational age and NDI in TTTS to weaken. Motor scores were significantly lower for children with severe cerebral injury. However, of the children with severe NDI, 59% had no evidence of severe cerebral injury. Fetal and postnatal magnetic resonance imaging (MRI) is not part of our routine care for preterm infants with or without TTTS. MRI is only performed when ultrasound scans show brain abnormalities. Some recent studies have shown the added value of MRI over ultrasound in the detection of cerebral injury in the context of TTTS [28,29]. However, the evidence is insufficient to confirm a predictive value of fetal MRI for long-term NDI, as this was never studied in large groups. Our findings emphasize the importance of long-term follow-up for all TTTS survivors, not just the ones born before 32 weeks of gestation or those with severe cerebral injury on ultrasound, as is the current practice in some fetal therapy centers around the world [30].

Strengths of this study are the very large group of included TTTS twins, as well as the use of a standardized psychometric instrument. Finally, every known TTTS pregnancy in the Netherlands during the study periods was treated in the LUMC. Consequently, there is no selection in terms of ethnicity, socioeconomic status or other factors that may be influenced by geographical differences. A limitation of this study is that survivors were assessed at two years of age, and testing at this young age only partially predicts NDI at a later age [31]. Therefore, it is important that follow-up for these children is continued until—at least—school age. Another limitation is that we experienced a slightly higher loss to follow-up rate than we did in our previous studies, with 15% of children lost to follow-up in the new cohort.

## 5. Conclusions

After a considerable decrease in NDI and an improvement in disease-free survival in the first decade of our laser program, this improvement now seems to have reached a plateau. With the incidence of severe NDI now being low at 3%, further improvement of disease-free survival may especially be gained by attempting to lower the incidence of immature and premature birth, because of the concomitant risks of low birth weight and cerebral injury, which affect long-term neurodevelopmental outcome, and death. Our data show the importance of assuring long-term follow-up for all TTTS survivors at least until school age, regardless of their gestational age at birth or the presence of severe cerebral injury on ultrasound scans.

## Figures and Tables

**Figure 1 jcm-08-01226-f001:**
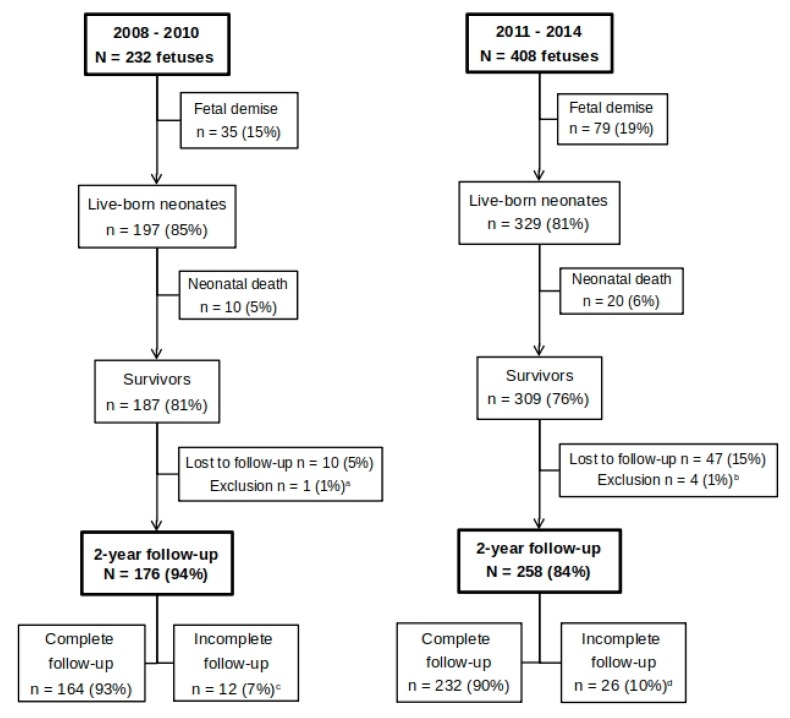
Flow chart of study population. ^a^ infantile Tay-Sachs disease; ^b^ n = 2: neurofibromatosis type 1, n = 2: familial severe hearing loss; ^c^ all 176 children were assessed by a pediatrician and physiotherapist; n = 5: only cognitive subtest of Bayley-III completed, no motor score; n = 7: neither cognitive nor motor subtests of the Bayley-III completed; ^d^ all 258 children were assessed by a pediatrician and physiotherapist; n = 8: only cognitive subtest of Bayley-III completed, no motor score; n = 2: only motor subtests of Bayley-III completed, no cognitive score; n = 16: neither cognitive nor motor subtests of the Bayley-III completed.

**Table 1 jcm-08-01226-t001:** Baseline characteristics of long-term TTTS survivors included for follow-up.

	Previous Cohort: 2008–2010 N = 176	New Cohort: 2011–2014 N = 258	*p*
Donor	90 (51)	130 (50)	0.685
Gestational age at laser (weeks)	20.3 ± 3.4	20.0 ± 3.4	0.663
TTTS stage—median (range) I II III IV	3 (1–4) 24 (14) 48 (27) 102 (58) 2 (1)	3 (1–4) 35 (14) 89 (34) 127 (49) 7 (3)	0.515
Post-laser TAPS or recurrent TTTS	20/176 (11)	25/258 (10)	0.523
Fetal demise of co-twin	18 (10)	29 (11)	0.771
Gestational age at birth (weeks) >37 weeks 33–36 weeks 26–32 weeks 24–25 weeks	32.4 ± 3.4 16 (9) 74 (42) 86 (49) 0	32.9 ± 3.1 21 (8) 116 (45) 112 (43) 9 (3)	0.516
Birth weight (grams)	1771 ± 596	1826 ± 610	0.717
Small for gestational age ^a^	16 (9)	22 (9)	0.835
Severe cerebral injury ^b^	9 (5)	11 (4)	0.681
Female	95 (54)	127 (49)	0.353

Data are presented as n (%), mean ± standard deviation or median (range). ^a^ Birth weight <p10 for gestational age according to the Netherlands Perinatal Registry (PRN) 2007. ^b^ Severe cerebral injury: at least one of the following findings on cerebral ultrasound: intraventricular hemorrhage (IVH) ≥ grade III, periventricular leukomalacia (PVL) ≥ grade II, ventricular dilatation ≥97th percentile, porencephalic cyst, arterial infarction, venous hemorrhagic infarction, or other severe cerebral lesions associated with adverse neurological outcome.

**Table 2 jcm-08-01226-t002:** Neurodevelopmental impairment in survivors included for follow-up.

	Previous Cohort: 2008–2010 N = 176	New Cohort: 2011–2014 N = 258	*p*
Severe NDI ^a^	10/169 (6)	7/241 (3)	0.189
Disease-free survival ^b^	162/215 (75%)	241/340 (71%)	0.263
Cerebral Palsy Cerebral Palsy grade I Cerebral Palsy grade II–V	5/176 (3) 1 (1) 4 (2)	4/258 (2) 3 (1) 1 (0.4)	0.356
Cognitive composite score	101 ± 14	99 ± 13	0.220
Cognitive composite score < −2SD	5/169 (3)	1/240 (0.4)	0.097
Motor composite score	102 ± 15	97 ± 14	0.003
Motor composite score < −2SD	5/164 (3)	6/234 (3)	0.723

N, number; SD, standard deviation. Data are expressed as n/N (%) or mean ± SD. ^a^ Severe neurodevelopmental impairment (NDI) included any of the following: Cerebral Palsy GMFCS > I, cognitive development <−2SD, motor development <−2SD, bilateral deafness or blindness. ^b^ Disease-free survival: the number of children without NDI at follow-up divided by the total number of included fetuses for whom the outcome was known (outcome either fetal/neonatal death or the presence or absence of severe NDI.

**Table 3 jcm-08-01226-t003:** Association of potential risk factors with cognitive and motor composite scores.

Risk Factor	Cognitive Composite Score Univariate		Motor Composite Score Univariate	
	B (95% CI)	*p*	B (95% CI)	*p*
Donor	−1.10 (−2.46–0.27)	0.115	1.29 (−0.67–3.25)	0.196
TTTS stage	0.10 (−2.108–2.30)	0.928	0.51 (−1.64–2.66)	0.642
GA at laser	−0.27 (−0.77–0.23)	0.291	−0.13 (−0.66–0.40)	0.634
Incomplete laser	−2.96 (−8.20–2.28)	0.268	−4.65 (−9.69–0.39)	0.071
Fetal demise co-twin	3.17 (−1.02–7.36)	0.138	1.89 (−2.71–6.48)	0.421
GA at birth	0.49 (−0.01–0.99)	0.056	−0.10 (−0.64–0.43)	0.707
Birth weight ^a^	0.41 (0.18–0.64)	0.000	−0.79 (−0.32–0.16)	0.521
Growth restriction ^b^	−5.67 (−9.35 to −1.99)	0.003	−0.47 (−4.52–3.58)	0.822
Severe cerebral injury	−2.11 (−8.96–4.73)	0.545	−14.10 (−25.04 to −3.16)	0.012

B, regression coefficient; CI, confidence interval; GA, gestational age. ^a^ Birth weight in grams/100. ^b^ Birth weight <10th percentile for gestational age according to the Netherlands Perinatal Registry (PRN) 2007.

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
