# Peer review of "Long-Term Neurodevelopmental Outcome in Twin-to-Twin Transfusion Syndrome: Is there still Room for Improvement?"

_jcm, 2019, doi:10.3390/jcm8081226_

Round 1

Reviewer 1 Report

Abstract: Please reorder the 955 CI in line 25 and use a coma instead of a dash because the CI is negative. It should read as 95: CI -3.16, -25.04.

Introduction: It is not clear what is novel about the study. The authors state that there are two different cohorts, but should clarify what makes these groups worth comparing. Was there a different technique used in the previous cohort versus the newest one? Please clearly state the purpose.

Section2. Experimental Section: I do not understand why TTTS cases that were not treated were excluded from this analysis. Please provide further detail as to why this occurred as well as why other groups were excluded (2nd paragraph of this section).

Page 72: Minor error: this is a sequence, so it should end with “…, and neonatal death.”

I appreciated that the authors clearly explained how results from the US version of the Bayley’s used in the first cohort were adjusted to coincide with new scores using the Dutch version of the test.

Flow Chart: 3 levels down, the chart shows how many were lost to follow-up, and how many were excluded; however, there is no indication of why these (1 and 4 participants, respectively) were excluded. Please add an additional arrow with more information on why these were excluded, or provide a superscript and a footnote indicating why these were excluded.

Section 3.3. Risk factors, line 216: What is meant by “>p10”? Is that a percentile of some kind? Please clarify this.

The authors indicate the need for long-term follow-up twice on Page 8 (lines 259 and 278), then again on Page 9, line 286-287. It is repetitive. Please consider deleting the previous two suggestions in lines 259 and 278, then saving it for your final conclusion.

Reviewer 2 Report

In this study the authors compared the incidence of severe neurodevelopmental impairment (NDI) in a recent cohort of TTTS survivors treated with laser between 2011-2014 with a previous cohort treated between 2008-2010.  They found no significant difference in survival without severe impairment between the two cohorts. Low birth weight, SGA and cerebral injury were found to be risk factors for NDI but surprisingly gestational age at delivery beyond 32 weeks did not exclude severe NDI. The topic of this paper is of great importance and includes a relatively large cohort that underwent a meticulous developmental assessment providing important data that will be helpful both in management and prenatal counseling of TTTS patients.

A few comments should be addressed:

1.       In how many patients in each cohort the Solomon technique was used? It seems surprising that the rate of past-laser TAPS was similar in both groups, since most cases in the recent cohort were done after the Solomon trial was published. Moreover, a rate of 12% for TAPS or recurrence seems high if Solomon technique was used.  The authors should address this point

2.       Was fetal brain MRI done in some of the patients? If Fetal MRI was performed, was there any correlation with severe NDI? Especially in those 10 cases with severe NDI, who had no evidence of cerebral injury on prenatal ultrasound. The role of fetal MRI in predicting NDI and its routine use in TTTS patients should be discussed, particularly since 53% of cases in their cohort with NDI had normal prenatal cranial ultrasound.

3.       If available, data regarding the degree of discordancy at laser and its association with the cognitive and motor composite scores should be added

4.       The authors should discuss and speculate why no association was found in this study between GA at birth and neurodevelopmental outcome, which is in contrast to previous findings.

5.       The authors should discuss potential mechanisms leading to NDI, as prematurity seems not to play a major role according to their data. Is it related to ischemic brain injury prior to the laser procedure?  Is it related to brain injury at the time of the procedure?
